# The Role of Homogeneous Waiting Group Criteria in Patient Referrals: Views of General Practitioners and Specialists in South Tyrol, Italy

**DOI:** 10.3390/healthcare12100985

**Published:** 2024-05-10

**Authors:** Giuliano Piccoliori, Christian J. Wiedermann, Verena Barbieri, Adolf Engl

**Affiliations:** 1Institute of General Practice and Public Health, Claudiana—College of Health Professions, 39100 Bolzano, Italyverena.barbieri@am-mg.claudiana.bz.it (V.B.); adolf.engl@am-mg.claudiana.bz.it (A.E.); 2Department of Public Health, Medical Decision Making and Health Technology Assessment, University of Health Sciences, Medical Informatics and Technology—Tyrol, 6060 Hall, Austria

**Keywords:** healthcare delivery, patient referral, homogeneous waiting group criteria, general practitioners, hospital physicians, relational coordination

## Abstract

Homogeneous waiting group (HWG) criteria are central to the patient referral process, guiding primary care physicians and hospitalists in directing patient care to specialists. This cross-sectional observational study, conducted in South Tyrol, Italy, in 2023, aimed to assess the implementation and impact of HWG criteria on healthcare from the perspective of general practitioners and hospital physicians. A questionnaire was developed to gain knowledge about referral practices as perceived by general practitioners and specialists. The survey included 313 participants (82 general practitioners and 231 hospital physicians) and was designed to capture a range of factors influencing the application of HWG criteria, including communication and collaboration practices. The results showed moderate levels of familiarity with HWG criteria and opinions about the need for criteria refinement among hospitalists, indicating that further education and refinement of these criteria are warranted. Both general practitioners and hospital physicians expressed dissatisfaction with the current specialist referral system, highlighting the significant gaps in effective communication and collaboration. The survey also demonstrated the influence of patient demands and waiting times on referral practices, and the need for streamlined and accessible specialist care. This study highlights the need for improvement and adaptation of HWG criteria to better meet the needs of healthcare providers and patients in South Tyrol. By addressing the identified gaps in communication, collaboration, and education related to the HWG system, the efficiency, effectiveness, and patient-centeredness of the referral process can be improved, ultimately leading to better health outcomes.

## 1. Introduction

In the Italian National Health Service, akin to the publicly funded model of the English health system [1], the ‘Homogeneous Waiting Group’ (HWG) criteria—‘Raggruppamenti di Attesa Omegenei’ (RAO)—play an important role in managing patient referrals from general practitioners (GPs) to specialists. The HWG criteria, developed by the Italian Ministry of Health, serve as a systematic tool for managing patient referrals to specialist services. These criteria classify patient cases according to urgency and complexity in order to prioritize care efficiently and ensure that patients receive timely access to necessary specialist consultations. An example of prioritization of clinical indications under the HWG criteria is detailed in Appendix A. The HWG criteria are key tools for regulating access to specialist hospital outpatient services and optimizing the balance between healthcare supply and demand [2]. In practice, GPs assess patient conditions using the HWG criteria to determine the appropriate level of urgency for specialist referral. This ensures that patients with more critical needs are prioritized, optimizing the use of specialist resources and reducing unnecessary waiting times. These criteria serve not only as a means of managing patient flows but also as an ethical response to the challenge of prioritizing care based on medical needs rather than on a first-come, first-served basis [3]. Figure 1 provides a flowchart of the patient referral process, illustrating the application of the HWG criteria.

The implementation of the HWG system reflects a broader European trend toward structured referral systems aimed at reducing waiting times and improving the quality of care in the face of economic and public health challenges [4]. This approach is particularly important in Italy, given the regional disparities in healthcare and the urgent need to ensure uniform standards of care in the face of projections that healthcare costs will continue to outpace economic growth [5,6]. The coronavirus disease 2019 (COVID-19) crisis has underlined the importance of common criteria for establishing clinical priorities to avoid delays in diagnostic pathways for non-COVID-19 diseases such as cancer [7]. This highlights the relevance of HWG criteria in contemporary health policies and the need for efficient “demand management” tools in the healthcare system [8].

Variability in referral practices remains a significant challenge across healthcare systems. Studies such as those by John et al. [9] have highlighted the inconsistent use of referral guidelines for colorectal cancer in different general practices, highlighting a pervasive problem in the standardization of referral processes. Similarly, Husum et al. [10] have identified discrepancies in the recognition and application of referral criteria among healthcare providers for hip dysplasia, suggesting a widespread need for improved clarity and adherence to referral protocols. Adherence to established referral criteria significantly improves patient outcomes [11]. However, there remains a gap in comprehensive research focused on the implementation and effectiveness of HWG criteria in streamlining referral processes in general practice.

Studies are needed to address this gap by evaluating the use of HWG criteria in the referral process from the perspectives of GPs and specialists, typically HPs. By examining how these criteria are used, key factors influencing the effectiveness of the HWG system can be explored. Furthermore, the integration of technological solutions in the referral process, as explored by Kidney et al. [12] in their study on colorectal cancer, points to the potential to enhance HWG criteria through digital tools.

This study aimed to evaluate the application of HWG criteria in practice, particularly from the perspective of GPs, who play a key role in navigating these guidelines to make informed referral decisions [13,14]. In addition, we examined aspects such as satisfaction with communication and collaboration between GPs and HPs, intensity and channels of communication, accessibility, perceived frequency and appropriateness of referrals, waiting times, and adherence to priority criteria. The effectiveness of HWG criteria is inherently linked to the dynamics of interprofessional collaboration, particularly between GPs, who act as gatekeepers to specialist services, and HPs, who function as specialists in outpatient services, frequently part of the hospital system. As various individual and contextual factors influence GPs’ referral patterns [15], it is imperative to understand the complexities of this process.

It would be useful for future studies to explore patient perspectives to provide a more holistic view of the impact of HWG criteria on healthcare delivery. The current study focuses on the perspectives of GPs and HPs because they are the primary users of HWG criteria in the referral process. Understanding their experiences and challenges is critical to evaluate the effectiveness of these criteria. By examining the views of GPs and HPs through responses to a comprehensive questionnaire, this study seeks to identify gaps and areas for policy intervention that could improve the functionality of HWG criteria. This approach aims to improve the efficiency and equity of the Italian healthcare system, thereby addressing policymakers’ pressing concerns regarding the efficiency, effectiveness, and appropriateness of healthcare. The healthcare system in South Tyrol is uniquely structured, with bilingual service provision and a high degree of autonomy in healthcare management, making it an ideal context for studying the implementation of HWG criteria and their impact on patient referral processes [16].

The main research questions of this study are as follows: How effectively are the HWG criteria implemented by GPs and HPs in South Tyrol? What is the perceived impact of these criteria on referral practices, including aspects such as communication, collaboration, waiting times, and adherence to priority criteria? The main objectives are to evaluate the application and effectiveness of the HWG criteria in improving the referral process, to identify and analyze the factors influencing GP and HP referral behavior, and to propose actionable recommendations for policy interventions aimed at refining these criteria to ensure a more efficient and equitable healthcare system. Thus, this study seeks to fill significant gaps in our understanding of the functioning of the HWG and its broader implications for healthcare delivery in Italy.

## 2. Materials and Methods

### 2.1. Study Design and Setting

This survey on an HWG was conducted as part of a broader study on relational coordination within South Tyrol’s healthcare system. While relational coordination findings are detailed in a separate publication [17], this study focused exclusively on the exploration and analysis of an HWG. This distinction allows for a concentrated examination of HWG practices outside relational dynamics.

This observational, cross-sectional survey primarily focused on evaluating opinions and attitudes toward the application and effectiveness of the HWG criteria in South Tyrol’s public health service. Located in northern Italy, South Tyrol is an autonomous province with a diverse population of 532,616 as of 31 December 2022, encompassing language groups such as German (62.3%), Italian (23.4%), Ladin (4.1%), and others (10.2%) according to the 2011 census [18]. The region confronts a significant healthcare workforce shortage, which is more acute than in other parts of Italy [19], partly because of the requirement for bilingual healthcare professionals [20,21]. Compounding this challenge is the region’s relatively slow progression in healthcare digitalization [22,23].

While this study’s central aim was to assess the implementation of HWG criteria, it also examined relational coordination in communication and collaboration among healthcare providers and aspects of job satisfaction and retention. These additional components, which are integral to understanding broader healthcare dynamics in the region, are discussed in separate publications.

### 2.2. Participants

The study participants comprised physicians actively involved in healthcare provision within the South Tyrolean Health Agency (SABES–ASAA), including both GPs and HPs.

The recruitment process involved sending specific e-mail invitations to the target groups. While the SABES–ASAA was responsible for inviting its employed HPs, the Institute of General Practice and Public Health reached out to GPs who had contractual agreements with the Health Authority.

The survey was implemented using an anonymous digital format facilitated by SoSci Survey Software, version 3.2.46. Data collection took place from 28 August 2023 to 2 October 2023. To encourage a broad response and achieve maximum participation, follow-up reminders were dispatched at two-week intervals following the initial invitation. Participation in the survey was voluntary, and informed consent was obtained from all respondents. The survey’s intent, confidentiality protocols, and the participants’ right to withdraw at any stage were communicated. The collected demographic data included age, sex, professional experience, and other factors potentially affecting perceptions of an HWG.

### 2.3. Questionnaire

Gittell’s theory of relational coordination has a significant influence on the development of survey items [24]. This theory, which emphasizes the importance of communication and relationships in coordinating work, has guided the creation of questions that assess the quality of interaction and collaboration among healthcare professionals. The key aspects of this theory reflect communication effectiveness, shared goals, mutual respect, and problem solving. These categories provide an understanding of how healthcare professionals collaborate and coordinate patient care, making them highly relevant to surveys that focus on healthcare dynamics.

The questionnaire for GPs and HPs was designed to assess factors influencing the application of HWG criteria, including both personal and contextual elements (Appendix A). Overall, the questionnaire employed a combination of Likert-scale, multiple-choice, open-ended, and quantitative questions to gather data on various aspects of medical collaboration and referral practices. The completion of the items on relational coordination was mandatory.

#### 2.3.1. Questionnaire for General Practitioners

In the GP survey, the questionnaire covered several areas to capture their experiences and perspectives on HWG criteria and related healthcare practices.

Impact of HWG Criteria: the survey explored perspectives on how HWG criteria affect waiting times, their agreement with these criteria and accessibility, and their adherence to the criteria in prescriptions (Items 7–10).Non-Compliance with HWG Criteria: Participants were asked about common reasons for not adhering to HWG criteria, such as long waits for normal appointments and clinical urgency not matching HWG criteria (Item 11). An open-ended question allowed GPs to specify which disciplines or areas required adjustments in priority criteria (Item 12).Collaboration and Communication: The questionnaire assessed GPs’ satisfaction with their collaboration and communication with HPs (Item 1). It also evaluated the frequency of phone and e-mail contact for patient-related clarifications (Items 2–3).Referral Practices: the questions focused on the frequency of avoiding or expediting specialist referrals through phone contact (Item 4) and the estimated percentage of acute consultation cases requiring specialist referral (Item 5).Influencing Factors for Referral: participants were asked to identify the factors that influenced their decision to refer patients to specialists, including the severity of symptoms, urgency of treatment, unclear diagnosis, patient requests, limited consultation time, and other factors (Item 6).Adherence to Expected Waiting Times and Specialties with Adherence Issues: these items addressed the extent of adherence to expected waiting times based on priority criteria and specialties with particular adherence issues to waiting times (Items 13–14).Opinions and Actions Related to Non-Urgent Appointments and Delayed Priority Specialist Visits: participants provided their views on waiting times for non-urgent appointments and the actions they took when a patient required a delayed-priority specialist visit (Items 15–16).

#### 2.3.2. Questionnaire for Hospital Physicians

For the HPs, the questionnaire was structured similarly to address various factors relevant to HWG criteria.

8.Impact of HWG Criteria: the questionnaire determined HPs’ level of awareness of current guidelines and HWG criteria for GP referrals (Item 16).9.Collaboration and Communication: queries assessed HPs’ satisfaction with collaboration and communication with GPs (Item 1), examined the frequency of phone contact with GPs (Item 2), sought suggestions for reducing waiting times (Item 3), evaluated the extent of connectivity between hospital departments and general medicine (Item 4), gauged the importance of a dedicated contact line for GPs (Item 5), and collected additional comments on improving collaboration (Items 6–8).10.Referral Practices: the survey evaluated the frequency with which HPs encountered urgent visits that they considered could have been non-urgent (Item 9), investigated the frequency of providing feedback to GPs regarding inappropriate referrals (Item 10), determined the necessity of open slots in clinics for direct GP contact (Item 11), assessed the clarity of clinical questions in GP referrals (Item 12), inquired about the frequency of inappropriate referrals due to patient requests (Item 13), explored the reasons for inappropriate referrals to hospital specialists (Item 14), and identified factors influencing GP decisions to refer patients to specialists (Item 15).11.Adherence to Expected Waiting Times and Specialties with Adherence Issues: this aspect sought opinions on the length of waiting times for normal, deferrable visits (Item 17).12.Opinions and Actions Related to Non-Urgent Appointments and Delayed Priority Specialist Visits: the survey captured beliefs about the potential improvement in referral criteria (Item 18), assessed satisfaction with the current referral system for specialists (Item 19), and explored attitudes toward alternative care methods such as telemedicine (Item 20).

### 2.4. Data Analysis

Data analysis included descriptive statistics to summarize the responses of both the GPs and HPs. Frequency distributions were used for categorical data to provide an overview of the response patterns across different variables. Means and standard deviations were calculated for continuous data, providing a quantitative summary of central tendencies and dispersion.

Differences between GPs and HPs were examined using comparative analyses. For categorical data, chi-square tests were used to compare proportions between the two groups and to identify statistically significant differences in categorical responses. For ordinal data, Mann–Whitney U tests were used to assess differences in rankings or scores, highlighting differences in opinions or practices between the two cohorts.

Open-ended responses were subjected to qualitative analysis. These responses were coded thematically and emerging themes were identified, providing deeper insights into participants’ perspectives and experiences that were not captured by quantitative measures.

Spearman’s rank correlation and Kramer’s V were also used to explore the interactions between different variables, providing insights into how different factors interact in shaping practices and opinions.

All statistical analyses were performed using IBM SPSS Statistics for Windows (version 25.0; IBM Corp., Armonk, NY, USA) to ensure a robust and reliable analytical framework).

## 3. Results

### 3.1. Survey Participation and Demographics of Respondents

During the recruitment phase of our survey, 1579 participants were invited, including 289 GPs and 1290 HPs (Figure 2). The access rate to the survey platform was significantly different between the two cohorts: 37% of the GPs and 24.1% of the HPs accessed it (*p* < 0.001). Non-completion after access was observed in 23.4% and 25.7% of participants, resulting in 82 evaluable questionnaires from GPs (28.4% of those invited) and 231 from hospitalists (17.9% of those invited), respectively. This resulted in an overall participation rate of 19.8%, and 313 evaluable questionnaires were collected for analysis.

A comparison of the demographic and professional characteristics of GPs and HPs who participated in the survey is presented in Table 1.

A comparison of the demographic and professional profiles of general practitioners and family physicians revealed a consistent age distribution in both groups, indicating a diverse age range among the health professionals. There is a significant difference in years of service, with GPs generally having fewer years of service than HPs do. Sex distribution was relatively balanced between the female and male participants in both cohorts, with no significant differences. A striking contrast can be seen in the linguistic background, where a higher percentage of GPs are German-speaking than HPs. This difference underlines the linguistic diversity within the health workforce in South Tyrol. Health district affiliations showed a broad representation across the region, with no significant differences.

### 3.2. Homogeneous Waiting Group Criteria and Their Impact

#### 3.2.1. Perceptions of Homogeneous Waiting Group Criteria among General Practitioners

The results of the GP survey on perceptions of HWG criteria are presented in Table 2. A diversity of opinions among respondents was found when evaluating the impact of HWG criteria on referrals. Approximately one-third believed that the criteria facilitated referrals or shortened waiting times, whereas almost 60% reported that referrals became more difficult, had longer waiting times, or had no noticeable effect as a result of the criteria. Thus, over half of the participants did not perceive an improvement in referrals, contrary to the intended goal of enhancing the referral process using the HWG criteria. Most GPs reported high levels of familiarity and agreement with the HWG criteria. The frequency of referrals from GPs to HPs varied. The distribution includes a range of percentages, with 19.5% of GPs referring less than 5% of cases, 34.1% of them referring 10%, and 23.2% of them referring 20%. In addition, 15.9% of GPs referred 30% of cases, and 3.7% reported other reasons. The average referral rate calculated from these responses was 13.8%. This varied distribution underscores the individualized approach that GPs take in determining when and how often to refer patients to HPs. On average, GPs reported that they believed they could meet the HWG criteria in approximately 70.1% of their referrals. However, it is worth noting that there is variability in individual responses, suggesting that GPs may have different levels of confidence in applying the HWG criteria to their referrals.

With regard to non-compliance with the HWG criteria, a significant number of GPs cited long waiting times for postponable visits as the most common reason. Other reasons included a mismatch between the clinical urgency and the HWG criteria, complaints that did not respond to treatment, an unclear clinical picture that did not meet the HWG criteria, and patient or family pressure.

Thus, the survey results reflect a complex picture of GP practices and attitudes toward referral criteria, with a spectrum of commitment and adherence to HWG guidelines.

#### 3.2.2. Awareness of Homogeneous Waiting Group Criteria among Hospital Physicians

The survey of HPs focused on their level of awareness of the HWG criteria for priority patient referrals by GPs. The question “How well informed are you about the current guidelines and criteria (HWG) for priority patient referrals by GPs?” was posed, with response options ranging from “very well” to “poorly” on a 1–4 Likert scale. The results showed that only 36.8% of hospitalists reported being “very well” or “well” informed about these criteria.

Regarding HPs’ opinions on whether refining the HWG referral priority criteria could reduce inappropriate referrals from GPs to specialists, the data showed that the majority of HPs (61.4%) answered affirmatively, with 17.7% answering “yes, definitely” and 43.7% answering “yes, probably.” On the other hand, 26.8% of HPs did not think that changing the criteria would make a significant difference (“no, probably not”), while a smaller fraction (7.8%) responded with a definitive “no, not at all”.

### 3.3. Communication and Collaboration

The survey explored the state of communication and collaboration between GPs and HPs. HPs were asked “How satisfied are you in general with the cooperation and communication between you and the GPs?”. In response, 52.9% indicated that they were either “very satisfied” or “satisfied”. Additionally, when HPs were questioned about “Can you reach the GPs by phone if anything is unclear?”, only 68% of participants reported being able to do so “frequently” or “occasionally”. Conversely, the GPs were asked a similar question: “How satisfied are you in general with the cooperation and communication between you and the hospital doctors?”. Here, 47.6% of the GPs reported feeling “very satisfied”. Regarding the ease of reaching their colleagues at the hospital by telephone, only 63.4% of GPs stated that they could do so “frequently” or “occasionally”.

The results indicated that about half of the physicians in both cohorts—GPs and HPs—were generally dissatisfied with their collaboration and communication with each other. In addition, approximately one-third of both groups reported that they could rarely reach their colleagues by phone for clarification, highlighting a significant gap in effective interdisciplinary communication.

In terms of how the two cohorts interacted and perceived each other’s patient management decisions, the survey explored the frequency and nature of the feedback regarding patient referrals. For HPs, responses to the question “How often do you give feedback to GPs about inappropriate referrals?” showed that only a small percentage (8.6%) reported giving such feedback “very often” or “often”, and about one-third reported giving it “occasionally”. A significant majority (60.2%) reported that they “rarely” or “never” provided feedback on inappropriate referrals. From the perspective of general practitioners, when asked “Have you received feedback from specialists in the past that a referral was inappropriate?”, a significant 30.4% of GPs reported receiving feedback that a referral was inappropriate “often” or “occasionally”.

#### 3.3.1. Perspectives of General Practitioners on Communication and Collaboration

When asked “How often do you e-mail HPs about patients?”, 58.5% of GPs reported doing so “frequently” or “occasionally”. Another item in the survey asked “How often do you succeed in avoiding a referral or reducing waiting time by making a telephone call?”. In response, 68.2% of GPs said they were able to do this “often” or “occasionally”, suggesting a considerable use of direct communication to speed up patient care.

#### 3.3.2. Perspectives of Hospital Physicians on Communication and Collaboration

Questions asked only of HPs included the following: “How well do you think your department in the hospital is networked with general practice?”. A total of 42% of respondents feel their department is either “very well” or “well” networked with general practice. Regarding the importance of having dedicated communication channels, when asked “How important do you consider ‘a fixed telephone number and/or e-mail address per department’, behind which there is a specialist who can be contacted daily by GPs?”, a significant 74.4% of respondents consider it “very important” or “important”. In response to the question “How would you rate the current working atmosphere and cooperate with colleagues from other disciplines in the hospital?”, 58.9% rated it as “very good” or “good”, indicating a positive collaborative environment within the hospital. On the other hand, only 6.5% rated it as “poor” or “very poor”, suggesting that problems with working atmosphere and collaboration are relatively rare.

### 3.4. Referral Practices

#### 3.4.1. Shared Referral Practice Perspectives of General Practitioners and Hospital Physicians

To explore opinions on factors influencing GPs’ decisions to refer patients to specialists, the survey asked each group the following question: “In your opinion, what factors most frequently influence the decision of primary care physicians to refer patients to specialists, and with what urgency?”. In their responses, 75.6% of the GPs cited the severity of symptoms (67.1%), urgency of treatment (74.4%), unclear diagnosis (31.7%), patient requests (4.9%), and time pressure (13.4%). The perceptions of HPs were significantly different; only 25.1% attributed referrals to the severity of symptoms, 20.2% to the urgency of treatment, and 51.6% to unclear diagnosis, but 59.6% attributed them to patient requests, 39.0% to practice time constraints, and 16.6% to other factors.

In response to the question “How satisfied are you overall with the current system of referring patients to specialists?”, the majority of both GPs and HPs (64.6% and 64.1%, respectively) expressed significant dissatisfaction with the current system of referring patients to specialists. 

Waiting time serves as an indicator of the effectiveness and efficiency of healthcare services. In response to the item “In your opinion, are waiting times for ‘normal’ deferrable visits generally too long?”, HPs feel that waiting times are “completely too long” or “mostly too long” (61.7% of the time), while GPs feel this way more strongly at 85.3%, suggesting that GPs may be more aware of or affected by the longer waiting times experienced by their patients.

Through the question “Do you support the expansion of alternative care modalities (e.g., telemedicine)?”, the survey revealed a difference in attitudes toward the use of alternative care modalities, such as telemedicine, with 54.1% of GPs and a significantly higher 73.6% of HPs supporting these innovations. This difference suggests a greater willingness among HPs to integrate newer technologies and practices into healthcare.

Both cohorts were asked the same question: “How necessary do you think it is to create free slots in specialist outpatient clinics for patients with special problems, which can be booked by the GP via direct telephone contact?”. A significant 76.8% of GPs responded that it was “very necessary” or “necessary” to create open slots in specialty clinics that could be accessed by direct telephone contact. Similarly, 67.5% of the HPs recognized the importance of this approach. Although this is slightly lower than the response from GPs, it still represents a significant majority of HPs who recognize the value of making specialist care more accessible and streamlined, particularly for patients requiring specialist care.

#### 3.4.2. Perspectives of General Practitioners on Referral Practice

As shown in Table 3, the frequency with which GPs refer patients to HPs in (re)acute consultations varies, with some doing so for a relatively small percentage of consultations and others more frequently. On average, GPs reported referring 26.1% of patients in (re-)acute consultations. The severity of symptoms, urgency of treatment, and unclear diagnoses are the main factors considered when deciding whether to refer patients to specialists, with patient requests and time pressure being less important.

Feedback from specialists suggests that some referrals are perceived as inappropriate, highlighting a potential area for improvement in referral decision-making. In terms of adherence to specified waiting times according to priority, a small percentage of GPs reported that these times were usually met, indicating challenges in providing timely patient care.

Specifically, longer waiting times are most commonly reported in specialties such as radiology, orthopedics, dermatology, ophthalmology, neurology, and cardiology. When faced with significant delays in priority visits or tests, GPs often refer patients to private specialists or structures, arrange earlier appointments directly with hospital doctors, or prescribe urgent visits to the emergency room.

The contracting of private structures with health services has been perceived by some GPs as partially successful in reducing waiting times. However, the problems associated with these referrals include a lower quality of services, increased bureaucracy, lack of follow-up, and other challenges.

#### 3.4.3. Referral Practice Perspectives of Hospital Physicians

The data on HPs’ perspectives on GP referral practices are presented in Table 4. HPs report different experiences with the urgency of referrals: some encounter a relatively small number of patients with priority or urgent referrals where a normal visit would suffice, while others see such cases frequently, suggesting a possible overuse of the “urgent” label. On average, GPs reported that HPs consider 34.9% of priority or urgent referrals inappropriate. When asked to rate the quality of the description of clinical issues underlying GP referrals, a modest proportion of HPs rated them as “very good” or “good”, with a larger proportion rating them as “fair”. 

A significant number of HPs frequently observed that patient demands lead to inappropriate specialist referrals, confirming that HPs are more likely than GPs to believe that patient expectations play a significant role in referral practice.

### 3.5. Correlations and Predictors of Hospital Physician Awareness and Views on Homogeneous Waiting Group Criteria Refinement

The mean scores for both familiarity of HPs with the HWG criteria and their opinion on the need for HWG criteria refinement were in the moderate range, with familiarity at a mean of 2.75 and the need for refinement at a mean of 2.26 on a 1–4 Likert scale. The medians for both criteria are 3.00 and 2.00, respectively, indicating a central tendency toward moderate values. The standard deviation for familiarity (0.067) and the need for refinement (0.057) suggest a moderate spread of responses around the mean, indicating varying perceptions among the HPs. Figure 3 shows these distributions, which are skewed toward the high end of the scale for familiarity and to the low end for agreement with the HWG criteria.

#### Correlation Analyses

A systematic presentation of the correlations between various sociodemographic factors and perceptions of referral practice, including levels of familiarity and agreement with the HWG criteria, among hospital physicians is shown in Table 5.

Table 5 shows the associations of sociodemographic factors and perceptions of referral practices with levels of familiarity and agreement with the HWG criteria among hospital-based physicians. Gender and medical district did not show significant associations with familiarity or agreement with the HWG criteria. Years in the hospital and years in practice were negatively associated with familiarity, meaning that shorter tenure was associated with higher familiarity with the HWG criteria, whereas no association was seen with agreement with the HWG criteria. 

Native language among HPs showed a significant association with both familiarity and agreement with the HWG criteria, with Kramer’s V association coefficients of 0.259 and 0.220, respectively, suggesting moderate associations between the variables. Scores on a 1–4 Likert scale (1 = yes, definitely; 2 = yes, probably; 3 = no, rather not; 4 = no, absolutely not) were used to further examine familiarity and agreement with the HWG criteria. German-speaking HPs had a mean familiarity score of 2.5 (SD = 0.09), with a median of 3. Italian-speaking physicians had a higher mean familiarity score of 2.9 (SD = 0.093), with a median of 3 (Mann–Whitney U test, *p* = 0.004). Regarding agreement with the HWG criteria, German-speaking HPs had a mean agreement score of 2.5 (SD = 0.09), with a median of 2, whereas Italian-speaking HPs had a lower mean agreement score of 2.11 (SD = 0.072), with a median of 2 (*p* = 0.002). These results suggest that German-speaking physicians tended to have slightly lower familiarity but higher agreement with the HWG criteria than their Italian-speaking counterparts did.

Increased familiarity with the HWG criteria correlated with lower overall satisfaction with specialist referral systems and lower overall satisfaction with general practitioner collaboration. Reduced familiarity with the HWG criteria was associated with increased perceptions of inappropriate referral frequency. Greater familiarity with the HWG criteria was significantly associated with more frequent communication of feedback to GPs about inappropriate referrals but not with overall interconnectedness with general practice.

### 3.6. Enhancing General Practitioner Communication and Collaboration

The GPs’ responses to improving communication and collaboration in the South Tyrolean Health Agency revealed key priorities. A significant majority of the results indicated concise summaries of organizational changes, easy access to administrative resources through a central link, and direct lines of communication with specialists. There is a strong preference for GP-led primary care services and, to a lesser extent, for regular meetings with primary care service directors to discuss issues. In addition, the need for reliable contact with hospital specialists is emphasized. These findings, detailed in Appendix A, highlight areas of potential improvement in the healthcare system.

## 4. Discussion

The results of the survey provide an overview of the current status of patient referrals from general to specialist care in South Tyrol. The observation of moderate familiarity with the HWG criteria among hospital physicians, coupled with the fact that almost two-thirds believe refining these criteria would improve referral appropriateness, suggests not only a need for further education and practice adaptation but also significant potential to optimize the criteria to enhance referral accuracy and efficiency. Both GPs and hospitalists expressed dissatisfaction with the current system of specialist referrals and identified significant gaps in effective communication and collaboration. Additionally, the survey highlighted the impact of patient demand on referral practices and the need for streamlined and accessible specialist care. These findings provide critical insights into areas that require attention to improve healthcare efficiency and collaboration in South Tyrol.

### 4.1. Homogeneous Waiting Group Criteria

GPs have different views on the impact of the HWG criteria, including mixed opinions on their impact on the ease of referral and waiting times, with the majority of them being skeptic. Although most GPs are familiar with and agree with the HWG criteria, their practice of prescribing priority visits or investigations varies widely. Non-compliance with the HWG criteria is often attributed to long waiting times and discrepancies between clinical urgency and the criteria. The results highlight a spectrum of commitment and adherence to HWG guidelines among GPs and illustrate the complexity of integrating these criteria into everyday practice [25].

Only a minority of HPs feel very well or well informed about the HWG criteria for priority patient referrals by primary care physicians, suggesting potential gaps in awareness or communication. Despite this, most HPs believe that refining the HWG criteria could reduce inappropriate GP referrals to specialists. However, a significant proportion of HPs remain skeptical about the impact of such changes, suggesting differing opinions regarding the effectiveness of the HWG criteria in current healthcare practice. This suggests a mixed understanding of the role and effectiveness of the HWG criteria among hospital staff.

On the one hand, the limited awareness of the HWG criteria among HPs may lead to the idea that GPs systematically fail to comply with these criteria. This raises questions about the basis of HPs’ criticism if their own understanding of the criteria is not comprehensive. On the other hand, HPs’ perceptions and experiences in dealing with GP referrals may lead them to believe that non-adherence occurs regardless of their level of awareness. The skepticism of HPs in South Tyrol regarding the adherence of GPs to the HWG criteria reflects a broader issue of organizational and communication challenges within the healthcare system. This scenario parallels findings from a study of integrated care for older patients, where adherence to guidelines varied across organizational levels [26]. Strengthening organizational structures and improving communication pathways may be critical to improving adherence to healthcare guidelines and protocols [27], suggesting a systemic rather than an individual problem with adherence to the HWG criteria.

The relationships between various sociodemographic factors and HPs’ perceptions of the HWG criteria were explored to identify the factors that influence familiarity and agreement with these criteria. The results showed that shorter hospital tenure was positively correlated with higher familiarity with the HWG criteria. The influence of native language emerged as a significant factor, with German-speaking HPs showing slightly lower familiarity but higher agreement with the HWG criteria compared to their Italian-speaking colleagues. These differences may reflect communication dynamics within a hospital setting influenced by language factors. Furthermore, the results showed that greater familiarity with the HWG criteria was associated with lower overall satisfaction with specialist referral systems and collaboration with general practitioners. Conversely, decreased familiarity was associated with increased perceptions of inappropriate referral frequencies. Unsurprisingly, physicians who were more familiar with the HWG criteria tended to provide more frequent feedback to general practitioners regarding inappropriate referrals, although this familiarity did not extend to the overall collaboration with general practitioners. These findings underscore the relationship between familiarity with the HWG criteria and the attitudes toward referral practices and communication with GPs.

### 4.2. Communication and Collaboration

A survey on communication between GPs and HPs revealed significant communication gaps. Only 53% of the HPs and 48% of the GPs reported satisfaction with interdisciplinary communication and collaboration. In addition, approximately one-third of both cohorts reported difficulty reaching each other by phone for clarification. These findings highlight the critical need for improved communication channels between GPs and HPs, similar to those reported in other healthcare settings [27], suggesting an impact on patient care coordination and healthcare efficiency. The inability of a significant proportion of both groups to reach their colleagues easily by telephone further underscores the need for more streamlined and effective communication channels. While these communication challenges are evident, their implications for the perceived effectiveness of the HWG criteria and their influence on referral practices warrant further exploration.

Approximately 30% of both cohorts reported an exchange of feedback regarding inappropriate referrals. This consistency in percentages between the two groups is significant and suggests a plausible level of communication regarding referral practice. This level of feedback, while not overwhelming, is substantial enough to suggest that there is active dialogue regarding the appropriateness of referrals. This reflects a system in which feedback, but perhaps not routine, is still a notable component of the referral process [28,29].

Expanding on the implications of observed communication gaps in the present study, the findings are consistent with the documented weak relational coordination between GPs and HPs [17]. This weak coordination, characterized by low frequency and timeliness of communication, critically undermines the effective implementation of the HWG criteria and directly impacts referral efficiency and overall healthcare delivery. The documented challenges underscore the need for targeted interventions to improve relational coordination.

### 4.3. Referral Practices

The survey directly probed the perceptions of GPs and HPs regarding several critical aspects of healthcare delivery. The data reflect moderate to high levels of agreement between GPs and HPs regarding the need for improved communication, better access to specialist appointments, and dissatisfaction with the current referral systems. The higher response rate among GPs regarding excessive waiting times suggests a particular concern from the primary care perspective regarding delays affecting patient care [30]. Despite general dissatisfaction with some aspects of the current system, there is notable openness to innovation, as seen in the responses to alternative methods of care, with GPs being significantly more receptive [31].

Comparing the responses to the question on the factors influencing GPs’ decisions to refer patients to specialists, interesting contrasts in perceptions emerged between the two groups. GPs appear to prioritize clinical factors such as symptom severity, urgency, and diagnostic clarity more than HPs perceive they do. Conversely, HPs believe that patient requests and time constraints play a greater role in GP referral decisions than GPs themselves report. This discrepancy highlights potential differences in the understanding or emphasis of referral practices between these two groups of healthcare professionals. Qualitative research in Switzerland has shown that GPs’ referral decisions are influenced by a complex interplay of not only these clinical factors, but also personal, emotional, and contextual elements [15]. This broader perspective suggests that the referral process is a multidimensional decision-making process, in which emotional and relational dynamics coexist with clinical considerations.

The variability in GP referral frequency, driven primarily by clinical factors such as symptom severity and diagnostic clarity, highlights a complex decision-making process. The perception of some referrals as inappropriate by specialists suggests a mismatch between GP referral decisions and specialist expectations. In addition, challenges in meeting specified waiting times, particularly in specialties with longer waiting times, can affect patient care. GPs’ strategies for managing these delays, including referrals to private facilities or direct arrangements with HPs while partially addressing waiting time issues, also raise concerns about service quality and increased bureaucracy. This scenario highlights the need for systemic improvements in waiting time management and coordination between general practitioners and specialists.

### 4.4. Proposals for Improvement

Several strategies are proposed to improve healthcare in South Tyrol:(i)Implementing educational programs for both GPs and HPs on the HWG criteria can promote a uniform understanding. Updating the HWG criteria in a person-centered manner can improve the referral system in the Italian healthcare system.(ii)Establishing reliable and accessible communication channels, including dedicated telephone and e-mail contact, is crucial. Tailoring these channels to meet the needs of both GPs and HPs can facilitate effective collaboration and coordination.(iii)Addressing workforce shortages through innovative referral management systems can improve the system’s efficiency. This approach, coupled with improved communication, can reduce waiting times and improve patient care.(iv)Policymakers must scrutinize the quality and effectiveness of contracted private providers to ensure that they meet health system standards and respond efficiently to patient needs.(v)Promoting better relational coordination can make nonclinical factors that influence referral decisions more transparent and acceptable. This requires a focus on building stronger relationships and an understanding among healthcare providers.

These strategies aim to create a more integrated, efficient, and patient-centered healthcare system in South Tyrol that will improve overall health outcomes.

### 4.5. Limitations

The limitations of this study include the moderate response rate, which may affect the generalizability of the findings. Despite the implementation of targeted outreach efforts, including invitations to all GPs working in the public health system and follow-up reminders to increase participation, the sample size of 82 GPs, although comparable to that of similar studies, remains a limitation. This context is important for interpreting the generalizability of our findings. In addition, we supplemented the quantitative responses with qualitative data to provide a richer, more nuanced understanding of GP perspectives on the HWG criteria, helping to offset some of the limitations of the quantitative approach.

As this study specifically examines the application of the HWG criteria within the South Tyrol healthcare system, it is important to recognize that the findings may reflect regional practices that are unique to this geographic and administrative context. The healthcare infrastructure, policy environment, and cultural factors in South Tyrol may contribute to specific challenges and outcomes that may not be directly replicable in other settings. Consequently, while the findings provide information on the functioning of the HWG criteria in this particular setting, caution should be exercised when generalizing these findings to other healthcare systems or cultural contexts.

Additionally, reliance on self-reported data might introduce response biases, and the limited availability of qualitative data limits deeper insight into communication and collaboration issues.

Furthermore, this study of patient referrals does not explore the perspectives of other healthcare stakeholders such as patients, nursing, or administrative staff, which might offer a more comprehensive understanding of healthcare dynamics.

## 5. Conclusions

This study provides insight into the effectiveness and challenges of current patient referral practices in South Tyrol. Key findings indicate that while the HWG criteria are a critical component of the referral process, there is a significant need for improved understanding and implementation by both HPs and GPs. Significant gaps in communication and collaboration have been identified, which not only impede the efficiency of referrals, but also affect the overall coordination of patient care.

The findings suggest that refining the HWG criteria and improving relationship coordination between healthcare providers could lead to significant improvements in healthcare delivery. By addressing these gaps, particularly through improved communication channels and targeted educational initiatives, the patient referral system in South Tyrol can be optimized. This would not only facilitate the referral process, but also ensure that patient care is timely, efficient, and focused on the best possible health outcomes.

Furthermore, this study highlights the importance of considering nonclinical factors, such as patient preferences, which significantly influence referral decisions. Incorporating these considerations can provide a more holistic approach to patient care and ensure that the referral system is not only efficient, but also equitable.

Future efforts should focus not only on technological improvements to facilitate better communication, but also on integrating comprehensive training programs for all healthcare providers involved in the referral process. These initiatives should aim to improve understanding of the HWG criteria, promote best practices in cross-disciplinary collaboration, and ultimately contribute to a more patient-centered healthcare system in South Tyrol.

## Figures and Tables

**Figure 1 healthcare-12-00985-f001:**
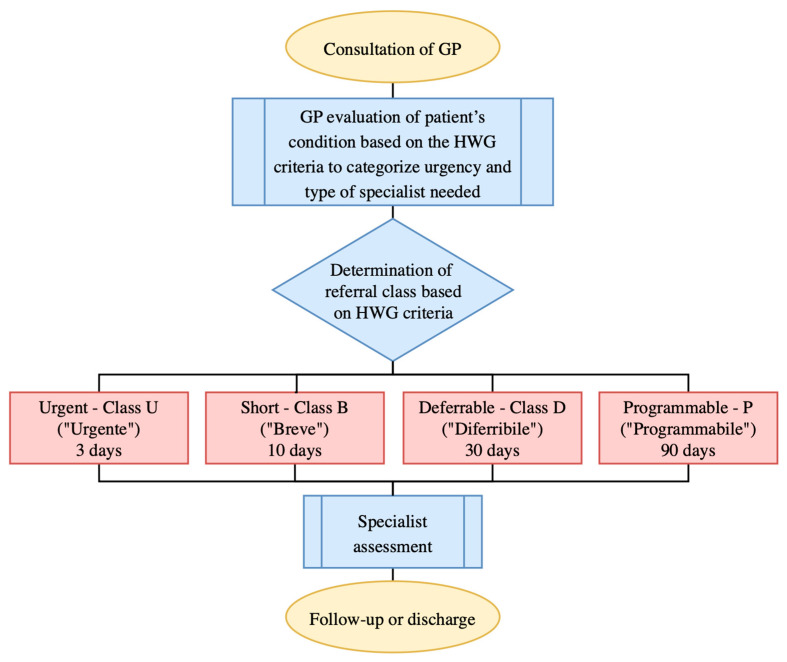
A flowchart of the patient referral process using the homogeneous waiting group criteria in Italy. The times given for first consultations are exemplary and come from the Veneto region. In the Autonomous Province of Bolzano, they are 1 day, 10 days, 30 days, and 120 days for classes U, B, D, and P, respectively. Abbreviations: GP, general practitioner; HWG, homogeneous waiting groups.

**Figure 2 healthcare-12-00985-f002:**
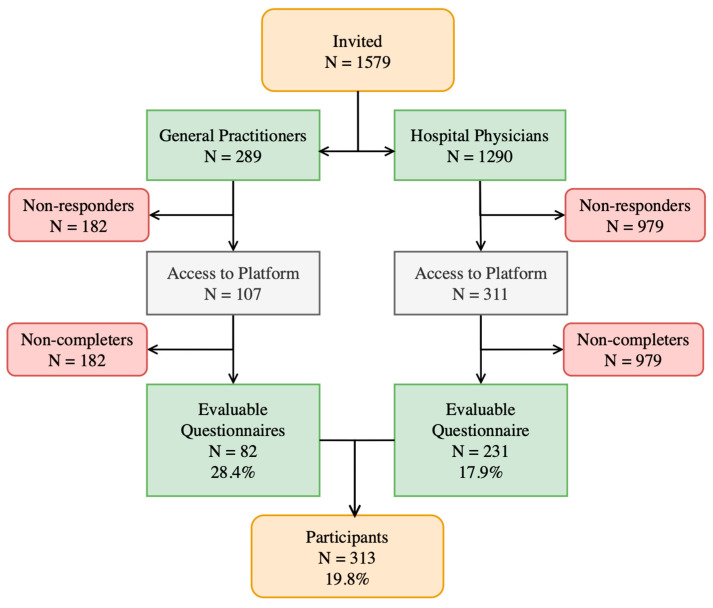
An overview of healthcare professional engagement in the homogeneous waiting group criteria study. The process of engaging general practitioners and hospital-based and out-patient service-based specialists with the survey is shown in the progression from being invited, to accessing the survey platform, and to their eventual participation or non-participation. Each node and connecting line is designed to provide a visual breakdown of the stages and response rates, illustrating the methodology of this study and the dynamics of participation.

**Figure 3 healthcare-12-00985-f003:**
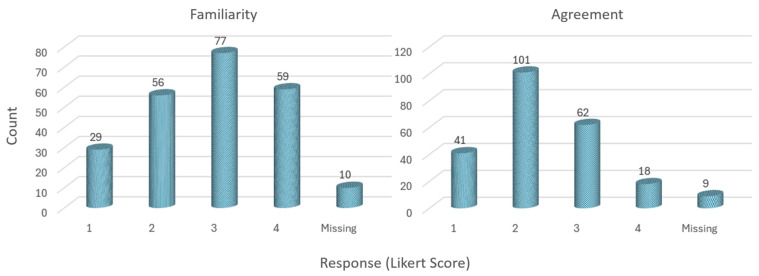
The distribution of familiarity with (left panel, N = 221) and opinions about the need to refine (right panel, N = 222) the homogeneous waiting group (HWG) criteria among hospital physicians on a 1–4 Likert scale (1 = yes, definitely; 2 = yes, probably; 3 = no, rather not; 4 = no, absolutely not).

**Table 1 healthcare-12-00985-t001:** Demographic and professional profile comparison between general practitioners and hospital physicians.

Variable	Total N = 313	GPs ^1^ N = 82	HPs ^1^ N = 231	*p*-Value
Age (M, SD years)	49 ± 9.7	50 ± 11.0	49 ± 8.9	n.s.
Years in service (M, SD)	17 ± 9.7	14 ± 10.5	18 ± 9.3	0.004
Gender (%)				n.s.
Female	48.2	47.6	48.5	
Male	40.6	42.7	39.8	
Missing	11.2	9.8	11.7	
German-speaking (%)	49.5	68.3	42.9	<0.001
Health district (%)				n.s.
Bolzano/Bozen	42.8	40.2	43.7	
Merano/Meran	21.7	18.3	22.9	
Bressanone/Brixen	15.0	19.5	13.4	
Brunico/Bruneck	12.8	12.2	13.0	
Missing	7.7	9.8	6.9	

^1^ Abbreviations: GPs, general practitioners; HPs, hospital physicians; M, mean; SD, standard deviation.

**Table 2 healthcare-12-00985-t002:** Perceptions of general practitioners regarding ‘homogeneous waiting group’ criteria.

Item	Response	Rate (%) ^1^
In your opinion, how have the HWG ^2^ criteria influenced the referrals?	Facilitated	29.3
	Made it more difficult	14.6
	Shortened waiting times	4.9
	Lengthened waiting times	22.0
	No effect on waiting times	22.0
If you are familiar with the HWG criteria or have access to them through your practice program, do you agree with them?	Yes, completely or yes, mostly	73.2
	No, rather not or no, not at all	22.0
How often do you estimate to prescribe HWG priority visits or examinations?	In less than 5% of referrals	19.5
	In 10% of referrals	34.1
	In 20% of referrals	23.2
	In 30% of referrals	15.9
	In ...% of referrals	3.7
How often do you think you can meet the HWG criteria in your referrals?	In 90% of referrals	42.7
	In 70% of referrals	32.9
	In 50% of referrals	14.6
	In less than 30% of referrals	6.1
What is the most common reason for non-compliance with the HWG criteria?	Waiting times too long for a deferrable visit	72.0
	Clinical urgency does not meet HWG criteria	43.9
	Complaints not responding to therapy	19.5
	Unclear picture not listed under HWG criteria	31.7
	Pressure/fear on part of patient or relatives	15.9
	Other	9.8

^1^ Missing and/or intermediate response option answers not shown. ^2^ Abbreviation: HWG, homogeneous waiting group.

**Table 3 healthcare-12-00985-t003:** General practitioner perspectives on referral practices.

Question	Response	Rate (%) ^1^
How often do you estimate that you refer your patients to a hospital doctor in (re-)acute counseling occasions?	In 10% of consultations	39
	In 30% of consultations	35.4
	In 50% of consultations	17.1
	In 80% of consultations	4.9
Are the waiting times specified according to priority (group ‘B’, 10 days; ‘D’, 30 days) usually observed?	Yes, completely or yes, for the most part	11.0
	No, rather not or no, not at all	85.3
If the waiting times specified according to priority are not observed, especially in which specialist disciplines? (open question)	Radiology	30
	Orthopedics	29
	Dermatology	20
	Ophthalmology	9
	Neurology	7
	Cardiology	7
	All/many	10
If, in your opinion, the patient needs a priority (group B, within 10 days) specialist visit/examination and it takes a significantly longer time to get it, how do you proceed?	You recommend the patient to go to a private specialist or structure	45.1
	You try to arrange an earlier appointment directly with a hospital doctor	23.2
	You prescribe an urgent visit for the patient and send them to the emergency room	23.2
	You recommend the patient to go directly to emergency room without your referral	0
Has the contracting of many private structures with the health service shortened the waiting times?	Yes, completely or yes, mostly	40.2
	No, not really or no, not at all	53.7
What problems have you encountered with referrals to private, contracted structures?	None	7.3
	Lower quality of services provided	43.9
	More bureaucratic effort (prescription of operations and follow-up treatments by the general practitioner, etc.)	74.4
	No aftercare	35.4
	Other	22.0

^1^ Missing and/or intermediate response option answers not shown; for open questions, response frequencies are given.

**Table 4 healthcare-12-00985-t004:** Hospital physician perspectives on referral practices.

Item	Response	Rate (%) ^1^
In your work, how often do you encounter patients with a priority or urgent referral from a general practitioner for whom, in your opinion, a “normal”, deferrable visit would have been indicated?	In less than 10% of priority or urgent visits	20.3
	In 10 to 30% of priority or urgent visits	26.0
	In 30 to 60% of priority or urgent visits	28.1
	In more than 60% of priority or urgent visits	20.3
How would you rate the indication of the clinical questions of GP referrals?	Very good or good	19.0
	Fair	44.6
How often do you have the impression that patients receive inappropriate specialist referrals due to patient demands?	Very often or frequently	61.5
	Occasionally or never	32.5
In your opinion, what are the most common reasons for inappropriate specialist referrals? (open question)	Patient’s pressure	17.7
	Missing knowledge/experience	10.8
	Lack of time	10.8
	Unclear diagnosis/lack of diagnostic tools	10.0
	Long waiting times	7.8
	Missing anamnesis/no personal examination	6.1
	Passing on responsibility	3.0
In your opinion, which factors most frequently influence the decision of general practitioners to refer patients to specialists and with what urgency? (multiple answers possible)	Severity of the symptoms	25.1
	Urgency of the treatment	20.2
	Unclear diagnosis	51.6
	Patient’s request	59.6
	Time pressure in the practice	39.0
	Other	16.6

^1^ No indication of missing answers. Abbreviation: GP, general practitioner.

**Table 5 healthcare-12-00985-t005:** Bivariate correlations of sociodemographic factors and perceptions of referral practice with levels of familiarity and agreement with HWG criteria among hospital physicians.

Variables	Correlation (ρ) ^1^			
	Familiarity (N = 221)	*p*-Value	Agreement (N = 222)	*p*-Value ^1^
Age	---	n.s.	---	n.s.
Gender	---	n.s.	---	n.s.
Medical district	---	n.s.	---	n.s.
Years of work in the hospital	−0.152	0.004	---	n.s.
Years of work in the field	−0.106	0.04	---	n.s.
Native language	0.259	0.002	0.220	0.013
Perception of inappropriate referral frequency	−0.260	<0.001	---	n.s.
Interconnectedness with general medicine	---	n.s.	---	n.s.
Overall satisfaction with specialist referral system	−0.145	0.032	---	n.s.
Frequency of feedback to GPs on inappropriate referrals	0.230	0.001	---	n.s.
Overall satisfaction with GP cooperation/communication	−0.167	0.013	---	n.s.

^1^ Spearman’s rank correlations, including gender, medical district, and native language, where we used Kramer’s V; correlation coefficients given only as significant *p*-values.

## Data Availability

The data presented in this study are available upon request from the corresponding author. The data are not publicly available because the survey had politically sensitive content.

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
