# Peer review of "The Role of Homogeneous Waiting Group Criteria in Patient Referrals: Views of General Practitioners and Specialists in South Tyrol, Italy"

_healthcare, 2024, doi:10.3390/healthcare12100985_

Round 1
Reviewer 1 Report
Comments and Suggestions for Authors
The review comment lies between rejection and major revision. The authors need to improve the manuscript comprehensively and fundamentally. The detailed comments are as follows:
1. The HWG criteria should be clearly explained. What's it? How it is used to facilitate the patient referral. Why only study HWG from the perspectives of HP and GP? Why not analyze the impacts of HWG from the perspectives of patients? The authors can use some figures to illustrate the patient referral process and the role of HWG.
2. The authors should enrich the literature review section to highlight the research gap and their contributions.
3. Only 82 questionnaires are collected from the GPs. The sample size is relatively small and it can undermine the conclusions. The authors need to collect more feedback.
4. Figure 1 is not clear.
5. The employed methodology is very simple. What are the main contributions? Do the insights obtained from South Tyrol also hold for other places in the world?
6. Do the referral process and HWG apply to all kinds of patients? HPs of different departments can have different views towards the HWG. The authors should justify that analyzing all HPs's feedback together is sound. The authors should improve the statistical analysis.
Comments on the Quality of English Language
There are typos and grammar errors. The writing should be improved.
Reviewer 2 Report
Comments and Suggestions for Authors
The paper provides the results of a survey on the application of the Homogeneous Waiting Group (HWG) criteria, which were introduced to reduce outpatient waiting lists in Italy. The questionnaire was submitted to general practitioners (GPs) and hospital physicians (HPs) in South Tirol to analyze GPs’ and HPs’ perceived impact of HWG on waiting list reduction, the referral practices and level of compliance of HWG criteria, influencing factors as well as adherence to expected waiting times based on priority criteria and specialties. These questions were rightly coupled with items regarding collaboration and communication between primary and secondary care in the practice of referring outpatients to diagnostic and specialist services.
As reported by the authors, the limited number of questionnaire respondents (82 GPs and 231 HPs) as well as the focus on a relatively small geographic area may affect the generalizability of the results. However, the paper provides interesting insights on GPs and HPs different perspectives in the application of HWG and their communication practices. The paper is clear and well structured. Methods and materials are well described and consistent with the analysis. Descriptive statistics is supplemented by correlation analysis and results are comprehensively discussed.
HWG criteria are differently used in other countries. Therefore, it could be useful to add in the supplementary materials some examples of priority criteria particularly in those specialties in which the priority indications given by GPs are considered to be not fully met. This could help readers outside Italy to have a more precise idea of the criteria used in Italy to reduce waiting lists.
Reviewer 3 Report
Comments and Suggestions for Authors
Thank you for the opportunity to review the paper. Interesting study. This study provides insights into the current state of patient referral practices in South Tyrol.
In the abstract
The authors should complete methods, and I recommend add the questionnaire used in the study.
Introduction.
The authors should add the research questions and main objectives at the end of the introduction. Furthermore, you should translate, there are Italian terms into English.
Methods:
We should add details on the validation process of the questionnaire. The authors should add details on the questionnaire's validation process. For example, they could include information on pilot testing with a small sample group or expert review by healthcare professionals familiar with HWG criteria.
Discussion:
While the discussion touches upon various aspects of the survey findings, I recommended discussing the implications of communication gaps in relation to the perceived effectiveness of the HWG criteria or how referral practices are influenced by communication and collaboration challenges.
For example: Should you add this paragraph or similar after line 498: "A survey on communication between GPs and HPs revealed significant communication gaps. Only 53% of the HPs and 48% of the GPs reported satisfaction with interdisciplinary communication and collaboration. In addition, approximately one-third of both cohorts reported difficulty reaching each other by phone for clarification. These findings highlight the critical need for improved communication channels between GPs and HPs, similar to those reported in other healthcare settings [22], suggesting an impact on patient care coordination and healthcare efficiency. The inability of a significant proportion of both groups to reach their colleagues easily via telephone further underscores the need for more streamlined and effective communication channels. While these communication challenges are evident, their implications for the perceived effectiveness of the HWG criteria and their influence on referral practices warrant further exploration."
In the limitations section, note that your findings may be unique to your country.
I hope my comments may be of help to authors in their work.
Reviewer 4 Report
Comments and Suggestions for Authors
Thank you for sharing your manuscript on this very interesting topic. This study examines the role of Homogeneous Waiting Group (HWG) criteria in patient referral processes from general practitioners (GPs) to specialists on the healthcare system in South Tyrol, Italy. The authors in this study conduct a survey collecting 313 samples, including 82 GPs and 231 hospital physicians. This survey assesses several HWG criteria, including the effectiveness, communication, and collaboration in the system. I have offered some comments for your consideration.
1. Abstract:
Please clarify the important findings in the larger context of healthcare in South Tyrol.
2. Introduction:
(1) The introduction could provide a more detailed explanation of how this study contribute to the existing literature or address a specific gap in current research. Emphasizing the novelty or urgency of the research question could strengthen the study's justification.
(2) Ensure the literature review comprehensively covers relevant prior work and demonstrates how this study builds upon or diverges from existing research.
(3) Identifying any overlooked studies or theoretical frameworks that could enhance the study's depth should be considered.
(4) Provide additional background on "Homogeneous Waiting Group" and "Patient Referrals," exploring their interrelation in existing literature.
(5) Clarify why General Practitioners and Hospital Physicians were selected for the study, emphasizing the significance of their perspectives in this context.
(6) Please provide more detail regarding the specific importance of focusing on South Tyrol to emphasizing the region's unique.
3. Materials and Methods:
(1) Please provide clarification on any methodologies or criteria that readers who are not familiar with this topic may not immediately understand. More information on survey design or data analysis techniques might improve the study's comprehensibility and repeatability.
(2) Conducting validity and reliability analysis is essential to make sure the rationale of the questionnaire.
(3) Why were the specific aspects of the HWG criteria selected? Elaborate on the rationale behind the design
4.Discussion:
(1) Please offer deeper insights into the implications of the findings and show the suggestion to the patient referral systems in South Tyrol. Did you find any unexpected results? What is the possible justification behind any unexpected results you obtained? Are there any limitations in the context of the framework of this study?
5.Conclusions:
(1) please make sure that the study's major findings and implications are reflected in the conclusions.
(2) Also, please expand on how this contributes to the patient referral systems in South Tyrol.
Comments on the Quality of English Language
Extensive editing of English language required
Round 2
Reviewer 1 Report
Comments and Suggestions for Authors
The modifications and responses are not satisfactory. The authors claim that the sample size is small because this study focuses on a region, where the GP and HP numbers are limited. However, the authors mention that the results, which are drawn from limited data of one region, can be also applied to other regions. This leads to serious logic flaws. The authors should expand the sample size, which should, at least, cover HPs and GPs of serveral regions. With an increased sample size, the statistical analyses can be improved as well. Otherwise, the contributions of this study are undermined significantly.
Comments on the Quality of English LanguageNA
Author Response
We appreciate your concerns regarding the sample size and the generalizability of our findings. As the study has already been completed, increasing the sample size further at this stage is not feasible. We acknowledge that extending the sample size to include GPs and HPs from multiple regions would enhance the statistical robustness of our findings; however, this was beyond the scope of the initial study design due to resource constraints.
Regarding the applicability of our results to other regions, we agree with your point about the limitations of our current data. Therefore, we propose to remove the statement suggesting that "These efforts will serve as a model for other regions seeking to improve their healthcare systems through similar referral mechanisms." We believe this modification will more accurately reflect the scope of our study and address the potential overgeneralization of our results.
Reviewer 4 Report
Comments and Suggestions for Authors
Having examined the latest version of the manuscript and corrections undertaken by the authors, I believe that the authors have improved the manuscript significantly in line with the comments.
Comments on the Quality of English LanguageMinor editing of English language required
Author Response
Thank you very much for your feedback and the acknowledgment of the improvements made to the manuscript.